# Addressing Challenges: Adopting Blockchain Technology in the Pharmaceutical Industry for Enhanced Sustainability

## Tino Riedel

Research Team, Chair of Entrepreneurship and Technology Transfer, HHL Leipzig Graduate School of Management, Jahnallee 59, 04109 Leipzig, Germany; tino.riedel@hhl.de; Tel.: +49-151-4260-3248

**Abstract:** The growing importance of sustainability in organizational success, particularly in the pharmaceutical industry, underscores the need for leveraging technologies such as blockchain methods to enhance sustainability indicators across environmental, social, and economic pillars. This study aims to identify and understand the challenges hindering the adoption of blockchain technology in the pharmaceutical sector for improving sustainability performance, addressing two research topics: the specific challenges faced by blockchain adoption in this context and the interdependencies among these challenges. Employing a two-step approach, the study compiles challenges through a literature review, refines them via expert opinions, and establishes their interrelationships using methodologies like fuzzy interpretive structural modeling (FISM) and cross-impact matrix multiplication applied to classification (MICMAC). The research contributes to unraveling the complex relationships and dependencies within the system, providing a structured framework for improved decision making and strategic planning. It fills a literature gap as the first attempt to outline driving and dependent factors related to the challenges of adopting blockchain technology for sustainability enhancement in the pharmaceutical sector, offering insights that can significantly impact brand image, company perception, and consumer value.

**Keywords:** blockchain; fuzzy-ISM; MICMAC; pharmaceutical industry; sustainability

## 1. Introduction

Given the increasing importance of sustainability for organizational success, especially in the pharmaceutical industry, it is recommended that companies strategically integrate sustainability into their business strategies [1–3]. Understanding how to leverage sustainability can garner support from investors, regulators, and consumers [4]. Emphasizing the strategic management of sustainability and reputation is crucial for enhancing corporate value, mitigating risks, and ensuring long-term business success. Additionally, considering sustainability reporting as a tool to bolster corporate reputation and brand equity is advised [5].

The pharmaceutical industry is pivotal in advancing the United Nations' Sustainable Development Goals (SDGs), which outline essential societal, environmental, and economic transformations for a prosperous future [4,6]. Pharmaceutical companies use these goals as a framework for their sustainable development initiatives, including waste management (environmental), healthcare access (social), and medicine availability (economic) [7]. Progress within the pharmaceutical sector is vital for achieving the SDGs [8]. The symbiotic relationship between the SDGs and innovative strides in the pharmaceutical industry is evident across various domains, such as embracing the "green chemistry" approach, which reduces the carbon footprint, promotes responsible production and consumption, and combats climate change [9,10].

One approach to addressing sustainability challenges in the pharmaceutical industry involves digitalization. It plays a significant role in enhancing sustainability indicators across environmental, social, and economic pillars. Organizations which fail to invest in

digital strategies and enablers, such as blockchain technology, supporting a wide range of sustainability goals and metrics, risk a significant impact on brand, company image, and consumer value perception [3,11]. Significantly, blockchain integration is expected to improve sustainability goals and impact all industries, creating opportunities for enhancing business processes and building trust in regards to data sharing and records management in every sector [12]. Pharmaceutical organizations like AbbVie employ blockchain technology to enhance supply chain transparency and security. By verifying product authenticity, detecting fraud, and improving quality assurance, these initiatives align with efforts to promote sustainable healthcare practices [13,14]. Pfizer's similar project, "End-to-End In-Transit Visibility," further supports sustainability by providing stakeholders with centralized product information, thus aiding the fight against counterfeit drugs, a significant threat to public health and safety [15].

In pursuit of the SDGs, blockchain technology is a pivotal enabling method capable of fostering sustainable and secure solutions. With characteristics such as accountability, transparency, traceability, cyber-resilience, and enhanced operational efficiency in global partnerships, blockchain technology has the potential to significantly contribute to these objectives [15]. Consequently, multiple essential sustainability metrics within the pharmaceutical industry can be improved through blockchain adoption [16]. Despite its potential as a lever to help organizations enhance their sustainability performance, there are still obstacles to the widespread adoption of blockchain technology. Even as research on the challenges of adopting blockchain technology has been growing, previous studies have illuminated multiple obstacles to its adoption, despite stakeholders' recognition of its potential capabilities [17]. Certain studies focus on specific countries and industries, whereas others embrace a broader scope. Thus, in adopting blockchain technology, in addition to universally applicable challenges like awareness and understanding, there may also be industry-specific, as well as country-specific, challenges, such as specific laws and regulations [18–20]. Nevertheless, empirical investigation has yet to be conducted concerning the challenges of adopting blockchain technology to improve sustainability within the pharmaceutical industry.

Building upon previous theoretical frameworks that address the challenges of adopting blockchain technology, this study aims to identify the specific challenges pertinent to the pharmaceutical industry to enhance sustainability performance and understand the interdependencies among these challenges. Thus, this study tackles two research questions:

RQ1: What challenges does the adoption of blockchain technology face in its effort to enhance sustainability performance within the pharmaceutical industry?

RQ2: What connections exist among the challenges related to the adoption of blockchain technology to enhance sustainability performance in the pharmaceutical industry?

The research methodology primarily utilizes a two-step approach, incorporating a three-phase decision framework. First, a compilation of challenges related to adopting blockchain technology in the pharmaceutical industry was accomplished via a literature review. Subsequently, the list of challenges was refined through the use of expert opinions by evaluating each challenge's significance using a Likert scale. To finalize the list, an analysis was conducted using statistical methods, and the reliability of the selected challenges was verified using a statistical test within the Statistical Package for the Social Sciences software version 28 by calculating the importance index and the correlation index modified to test consistency (CIMTC). Ultimately, the interrelationship of these finalized challenges was established using the FISM and MICMAC methodologies.

As a result of this study, the challenges hindering and impeding the adoption of blockchain technology for enhancing sustainability performance are revealed. Among the numerous challenges, those that play a significant role in this context are identified. Additionally, various directly and indirectly related challenges are structured into a comprehensive systematic model. This resulting model presents the problem's structure using a carefully designed pattern. This approach identifies relationships among specific challenges, allowing for a more precise description of the situation than when each isolated

factor is considered. Thus, a structured framework for unraveling a system's complex relationships and dependencies, enhancing decision-making and strategic planning processes, is provided.

The study closes the current literature gap regarding the understanding of the dedicated challenges hindering pharmaceutical organizations from adopting blockchain technology to enhance their sustainability performance. Furthermore, it is the first attempt to outline the driving and dependent factors related to the challenges of adopting blockchain technology to improve sustainability in the pharmaceutical sector.

## 2. Literature

### 2.1. Impact of Blockchain and Sustainability

Blockchain technology, as a decentralized transaction and data management system, offers a range of capabilities [21]. Its appeal lies in its capacity to facilitate transparent data sharing, optimize business processes, reduce operating costs, enhance collaborative efficiency, and establish a system that does not require explicit trust incorporation [22]. Additionally, it enables innovative approaches to green production, including the monitoring and storing of data related to pollution and environmental degradation Real-time collection and analysis of green or low-carbon data results in prompt decision making [23]. These advancements present significant opportunities for progress in business, supply chain innovation, and sustainable development [24].

Widely recognized for its potential to enhance supply chain sustainability, blockchain technology advances security, accountability, and efficiency [25,26]. The blockchain philosophy, guided by principles of democracy and decentralization, contributes to establishing more equitable supply chains. It can be incorporated as a comprehensive strategy to achieve multifaceted objectives, such as supply chain mapping, sustainability, and integration [27]. Positioned as a potential sustainability-oriented innovation, blockchain technology entails intentional changes to an organization's philosophy, values, products, processes, or practices to generate and actualize social and environmental value alongside economic returns [28,29].

Furthermore, blockchain technology is advocated as a solution to current challenges, offering the potential to strengthen food security, mitigate fraud, ensure fair labor practices, and reduce waste and $CO_2$ emissions [21,30]. Its potential to support the circular economy by lowering transaction costs, improving performance and communication throughout the supply chain, safeguarding human rights, and enhancing healthcare patient confidentiality and well-being is noteworthy [31].

How blockchain technology addresses social, economic, and environmental challenges highlights how the technology aligns financial performance with sustainability objectives. The potency for generating shared value is evident in social and economic sustainability. The enthusiasm for blockchain technology is apparent in its current applications, offering new opportunities to enhance sustainability. Whether used as a tool, a mindset for sustainability, or both, blockchain technology can play a crucial role in overcoming challenges and improving the sustainability performance of organizations [32,33].

### 2.2. Sustainability and Blockchain Technology in the Pharmaceutical Industry

The decentralized nature of blockchain technology makes it more reliable, transparent, and traceable in business processes, where all transactions are consistent, stored, immutable, and distributed among all network nodes [34]. In the pharmaceutical industry, blockchain technology enables the achievement of the top five influential factors: monitoring, reliability, traceability, authorization, and real-time functionality [35].

The pharmaceutical supply chains differ significantly from typical supply chains, as any disruption in the pharmaceutical supply chain can have direct implications for patient well-being [36]. A potentially effective solution to tackle the pharmaceutical supply chain's significant challenges is blockchain technology, which offers traceability throughout the

product's lifecycle by linking, disseminating, and transmitting data within an organization. This aspect is particularly crucial for highly regulated pharmaceutical industries [37].

Furthermore, various vital participants, including manufacturers, wholesalers, and retailers, are engaged in the production, transportation, distribution, and sale of medications traversing a supply chain [38]. Thus, a worldwide system for global supply chains to swiftly alert people worldwide about the risks posed by substandard and falsified medical products is essential, since 50% of drugs consumed in developing countries are counterfeit [39,40]. As counterfeit medicine has become one of the world's most intricate and challenging issues, blockchain technology facilitates the monitoring of drugs at every stage of the supply chain [41,42].

The tracing feature of blockchain technology can also be leveraged regarding waste management, a central challenge for the pharmaceutical industry [43]. Waste management raises concern due to its potential threat to human and environmental health. Given the associated risks, pharmaceutical waste cannot be treated like regular waste, and it necessitates special handling, whether from a hospital, clinic, pharmacy, or private household [44]. Blockchain technology can enhance waste management in the pharmaceutical industry by ensuring data privacy, compliance, cost-effectiveness, and expeditious trash collection and disposal. Traditional paper records have been replaced with electronic medical records, which are more readily available, secure, and shareable. In the pharmaceutical industry, blockchain technology can guarantee data consistency and security through a single source of truth, reliable verification procedures, and tamper-proof transactions. The transparency and tamper-proof nature of blockchain technology can facilitate drug monitoring, eliminate fraud, and enhance supply security [45].

Utilizing blockchain technology in the pharmaceutical sector for sustainability improvement, while promising, comes with several potential drawbacks. There are concerns regarding the complexity and cost of integrating blockchain technology into existing infrastructure and processes. This can involve significant investment in technology and expertise and potential disruptions to operations during the implementation phase [45]. Furthermore, scalability remains challenging for blockchain networks, particularly in pharmaceutical industries with high transaction volumes. Ensuring that blockchain platforms can handle large-scale data processing efficiently is crucial for their effectiveness [46].

Additionally, regulatory compliance poses a significant hurdle, as pharmaceutical companies must navigate a complex landscape of regulations and standards. Ensuring that blockchain solutions comply with existing laws and industry regulations while addressing emerging regulatory frameworks specific to blockchain technology requires careful attention and resources [47]. While the adoption of blockchain technology holds significant promise for the sustainable development of the pharmaceutical industry, and despite the growing body of research on the challenges associated with its adoption, a comprehensive understanding of the potential challenges in adopting blockchain technology is still in the early stages [17].

### 2.3. Literature Review

Employing a systematic literature review (SLR) methodology, the existing body of literature was systematically explored. SLR enables a transparent and comprehensive search across multiple databases, ensuring the replicability and reproducibility of the available literature [48,49]. This widely adopted methodology, utilized by scholars such as Durach et al. (2017) [50], Sansone et al. (2017) [51], Wetzstein et al. (2016) [52], and Sangwa and Sangwan (2018) [53], aids in identifying, selecting, and reviewing relevant literature in the field of study. The literature selection process is displayed in Figure 1. The literature review used a Boolean search across several electronic databases such as *Springer*, *IEEE Xplore*, *Scopus*, *ScienceDirect*, *Web of Science*, *Emerald*, *EBSCO*, and *Taylor & Francis*. This search type was employed due to numerous interfaces within the search domain and its capability of limiting, expanding, and molding the results by incorporating additional elements into the search query [54,55]. The following search was performed and considered conference

papers, journal articles, and book chapters published in English: "challenges of blockchain adoption" OR "challenges to blockchain adoption" OR "barriers of blockchain adoption" OR "barriers to blockchain adoption". A total of 69 studies were identified which can be located in the Supplementary Materials (Table S1). Despite the widespread anticipation of blockchain's influence on all sectors, few studies have examined the challenges of adopting blockchain technology [56]. The SLR indicates the existence of several studies exploring the challenges related to the adoption of blockchain in *Taylor & Francis*. These studies cover perspectives specific to industries, as well as those of a more general nature. Additionally, the researchers utilize qualitative and quantitative approaches to identify these challenges. It is essential to highlight that only a small subset of studies delve into the interconnections among these challenges. Our SLR also emphasizes that prior empirical or quantitative assessments of the challenges associated with blockchain adoption are required, particularly in improving the pharmaceutical industry's sustainability performance.

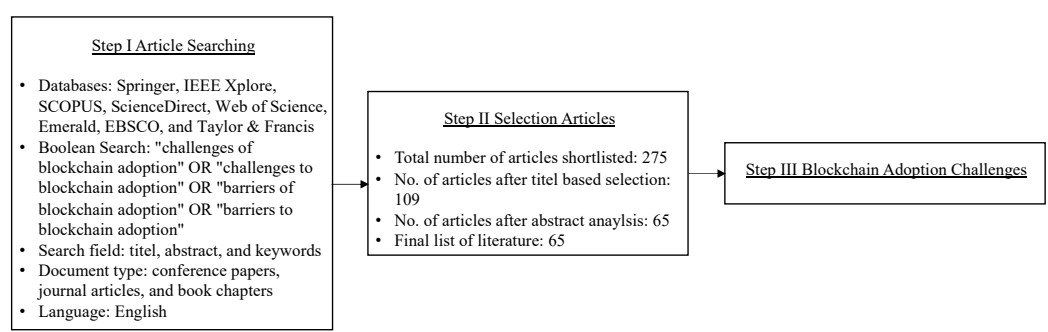

**Figure 1.** Literature selection process.

## 3. Methodology

A three-phase decision framework was employed to evaluate the obstacles associated with adopting blockchain technology to enhance sustainability in the pharmaceutical sector and to analyze the interrelationships among them, as depicted in Figure 2.

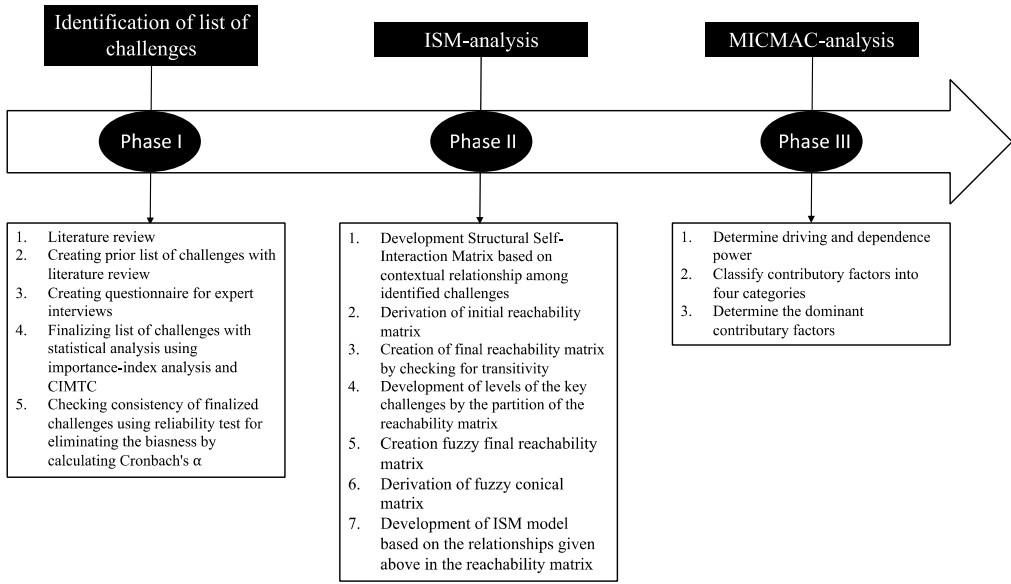

**Figure 2.** Three-phase decision framework.

In the initial phase, a comprehensive compilation of challenges related to adopting blockchain technology was discerned through an SLR and expert interviews. This compilation was then refined and finalized through statistical analysis. The reliability of the identified challenges was assessed using a statistical test within the Statistical Package

for the Social Sciences (SPSS) software version 28. Moving to the second phase, an FISM methodology was applied to establish connections among the identified challenges. Finally, in the third phase, an MICMAC analysis was conducted to categorize the challenges based on their driving and dependence power.

### 3.1. ISM Analysis

Various MCDM techniques have been applied in barrier studies, with the decision making trial and evaluation laboratory (DEMATEL), interpretive structural modeling (ISM), and analytical hierarchy process (AHP) methods being the most commonly used techniques for analyzing relationships between factors. Although AHP is widely used for simplicity, it may not effectively analyze complicated interdependencies among factors. DEMATEL and ISM, on the other hand, are favored for their ability to capture and represent interdependencies comprehensively [57,58]. ISM and DEMATEL are powerful structural modeling tools, offering a clear, hierarchical representation of the relationships among factors [59].

DEMATEL is primarily utilized for comprehending and illustrating the direction and strength of cause–effect relationships, both direct and indirect, among various criteria [60,61]. It enhances the understanding of complex issues and helps identify practical solutions within a hierarchical structure. DEMATEL is widely recognized and employed as one of the most effective models for visualizing and addressing intricate interconnections among various factors, capturing the intensity of the influence using a Likert scale (e.g., 0–4) [61].

In various cases, ISM organizes dissimilar and directly or indirectly related factors or components. This method relies on multiple independent expert opinions to determine how factors are interrelated, creating a structured model. It is a modeling approach that develops diagram models of complex factor interactions categorized into four possible hierarchies, enabling clear and precise inferences in the given context. ISM is a valuable tool for transforming unpredictable problems into well-defined models that can be effectively communicated, representing the relationships and overall structure through a digraph model. This technique has found applications in solving complex real-life problems across different industrial settings, such as analyzing barriers in reverse logistics and understanding the drivers of green supply chain management. Moreover, numerous researchers have adopted it as a methodology [62–64].

To demonstrate the intensity of relationships between variables, Saxena and Vrat adapted ISM in 1992, creating fuzzy interpretive structural modeling (FISM). While ISM primarily considers variables as interconnected, FISM delves deeper into the strength of these connections. FISM operates on the premise that relationships between variables exhibit variability. Unlike ISM, which employs precise and definite values, FISM acknowledges that certain aspects of elements cannot be assigned crisp, exact values. This is because the outcomes are contingent upon the preferences of decision-making experts. Therefore, fuzzy logic is considered a more practical approach for solving problems with inherent uncertainties, especially when multiple decision makers are involved [65].

One could argue that FISM offers more significant advantages compared to DEMATEL and ISM due to its ability to introduce rationality through the consideration of mutual relationships [66]. Furthermore, it provides a comprehensive approach to analyzing complex systems by accommodating ambiguity, incorporating subjective inputs, and combining qualitative and quantitative analysis. This flexibility allows for a thorough understanding of structural relationships and their relative importance, making it a valuable decision-support tool in strategy development and risk assessment [67]. In this paper, FISM is applied to enhance the rationality of the ISM model in evaluating the relationships among system units. The precise procedures are outlined as follows:

1. Identification of the challenges of the concerned approach.
2. Establishment of a contextual relationship among challenges through expert survey.

3. Creation of a structural self-interaction matrix (SSIM) based on the contextual relationships between variable pairs.
4. Development of a reachability matrix from the SSIM and checking for transitivity.
5. Partitioning of the final reachability matrix into different levels.
6. Creation of a fuzzy conical matrix.
7. Development of the FISM model, based on the relationships given above in the reachability matrix [68,69].

*3.2. MICMAC Analysis*

MICMAC is a method used for classification and examination, leveraging matrix multiplication properties [70]. It is employed to explore both the driving force and dependency of challenges [71]. MICMAC analysis aims to evaluate and classify variables based on influence and dependency, placing them into four groups: autonomous, dependent, linkage, and independent [72]. Its visual representations and flexibility make it an effective method for strategic decision making across diverse domains, optimizing resource allocation, and facilitating comparative analyses [73,74]. Characteristics associated with autonomous clusters exhibit weak influence and dependency. These attributes distinguish themselves within the model, as characterized by a sparse yet impactful network, and have minimal impact on the system. The absence of elements within autonomous clusters emphasizes the importance of all factors, necessitating comprehensive attention from practitioners [75]. The challenges categorized as dependent exhibit a relatively low driving force but demonstrate a notable reliance on other barriers [76]. Elements identified as independent possess limited influence but significant dependency, relying on external factors and deserving increased consideration. Elements showing substantial dependency may also concurrently exhibit linkage due to their strong influence [77].

## 4. Research and Data Analysis

*4.1. Data Collection*

MICMAC analysis of the final literature list, developed as outlined in Section 2.3, revealed 24 challenges related to blockchain adoption. The details, encompassing challenge titles, descriptions, and associated references, are comprehensively presented in Table 1.

**Table 1.** List of challenges and description of blockchain adoption.

| No. | Challenge | Description | References |
|---|---|---|---|
| 1 | High costs of blockchain investments | Organizations can incur significant costs, particularly during the initial adoption phase, as they may need to develop software, including encryption and tracking technologies, and invest in additional hardware to establish blockchain-based operating systems. Moreover, while current blockchain usage is free, as adoption increases, subscription fees might be introduced due to network saturation, as has occurred in the case of other technology adoption scenarios. | [8,17,20,23,78–107] |
| 2 | Lack of regulatory framework | If proper governmental regulations (data security and privacy laws) are not put in place, the widespread adoption of blockchain technology will remain hindered. | [8,17,20,68,86–91,93,99,100,103,105,108–116] |
| 3 | Lack of management support | Effective support from senior management is essential for the successful implementation of any sustainability initiative. This is especially true for the adoption of technology, where organizational leadership plays a crucial role. | [8,23,78,85,88,94,99–102,111,116–120] |
| 4 | Lack of security | Issues related to security can lead to data vulnerabilities, hacking risks, loss of confidentiality, reputation damage, regulatory non-compliance, and reluctance to adopt the technology. Collectively, these issues can hinder the adoption of blockchain technology in business settings. | [8,17,20,23,79,80,83,85,87–89,93–95,97,99,101–103,105,109–111,113,115,116,121–126] |

**Table 1.** *Cont.*

| No. | Challenge | Description | References |
|---|---|---|---|
| 5 | Difficulty in changing organizational culture | The adoption of blockchain technology results in the alteration or evolution of the existing organizational culture. Organizational culture encompasses norms related to the work environment and suitable conduct within the organization. | [23,79,94,100,110,113,119,122] |
| 6 | Lack of standardization and homogeneity | Standardization has the potential to enhance efficiency, particularly through features like smart contracts, and it specifies transaction structures, validation, and security within the blockchain network. Without established standards, corporate hesitation is unlikely to diminish, and standardization can also break down technical silos, aligning with the blockchain's core aim of dismantling information silos and enabling horizontal integration. | [20,85,88,89,92,95,96,101,106,107,122] |
| 7 | Degree of immutability | Immutability can create challenges when it comes to data accuracy, regulatory compliance, and adaptability since once data is recorded in a blockchain, it cannot be easily altered or deleted. While this is a key feature for security and trust, it can be problematic if errors are made or if there is a need to update information. The immutability feature can also pose legal and regulatory challenges. | [8,23,80,85,90,99,101–103,122,126] |
| 8 | Lack of interoperability | Interoperability refers to the capability of different blockchain networks or systems to communicate and share data seamlessly. When interoperability is lacking, efficient data exchange between networks is prevented due to fragmented landscapes, and a smooth transfer of assets and data across platforms is hindered. This siloed approach limits innovation, increases complexity, and can result in vendor lock-in, since one may be dependent on a single blockchain platform. | [17,20,23,68,80,81,86–90,92,94,95,99,101,102,106,108,113,115,122,124,125,127] |
| 9 | Lack of resources | The complexity of blockchain technology may demand a significant investment of time and resources for companies to become proficient, while the expenses associated with hiring blockchain experts can be exceptionally steep due to high demand in the field. | [20,23,90,92–95,99–101,103,108,111,113,118,122,124] |
| 10 | Lack of governmental support and regulations | Despite the increasing number of countries considering initiatives to embrace distributed ledger technology (DLT) and blockchains, tangible support from governments—such as incentives to adopt blockchain technology or laws that concern data sharing—remains limited and uncertain at this juncture. | [20,23,78,79,85,87,89,90,92–95,97,99,101–103,107–111,113,114,117,118,120–123,125] |
| 11 | Lack of market acceptance | The market's uncertain stance in regards to accepting the technology and its associated products acts as a deterrent for managers when contemplating the risk of such investments and leads to trust issues among the stakeholders | [8,17,19,20,23,26,85,90,92,95,96,98–103,105,106,109,112,116–118,122] |
| 12 | Lack of scalability | The restricted block size in blockchain technologies has led to scalability challenges, with Bitcoin's block size of 1 MB allowing just seven transactions per second. Scalability affects the ability to handle growing transactions, while slow speeds hinder real-time processing. These limitations can impede adoption in various sectors and discourage investment in blockchain projects. | [8,17,20,80,81,83,89,90,92,95,101–103,111,113,115,121–123,125] |
| 13 | Lack of maintenance and management | Without regular maintenance, blockchain networks experience slower transaction speeds and reduced efficiency, leading to performance degradation. Ignoring security updates exposes the blockchain to exploitation by malicious actors, compromising data security, which can lead to compatibility issues with newer systems. Furthermore, inadequate management can introduce errors, eroding the trust in the immutability of blockchain data and resulting in unexpected downtime, disrupting operations and leading to users losing trust in poorly maintained blockchains, impacting adoption. | [17,20,80–82,85,86,115,122,124] |

**Table 1.** *Cont.*

| No. | Challenge | Description | References |
|---|---|---|---|
| 14 | Lack of technology expertise and skills | Without a solid grasp of the intricacies of blockchain technology, there is a higher likelihood of poorly designed or executed blockchain projects, resulting in suboptimal outcomes leading to reluctance in regards to adopting blockchain solutions due to perceived risks. Furthermore, organizations may struggle to formulate effective strategies around blockchain integration without a clear understanding of how the technology aligns with their goals, including employee training. | [8,17,20,23,78–82,85,86,89–93,95,99,101,102,104,105,107,108,110,111,113,117–119,121–125] |
| 15 | Lack of data privacy | Since blockchain transactions are recorded on a public database accessible to anyone, it generates a setting that gives rise to privacy concerns regarding this technology. | [17,20,23,26,79,80,83,85,87–95,100–102,105,106,108,110,113,115,118,122,124–126] |
| 16 | Lack of validation | Due to limited piloting, insufficient validation could impede the adoption and utilization of blockchain technology. | [23,80,81,92,93,99,115,118,127] |
| 17 | Costs of latency | Given the inherent structure of the architecture, which requires the synchronization of all blocks within the chain for any new additions, this process could be resource-intensive, particularly in the case of extensive blockchains. This computational demand might pose a potential hindrance to implementation. | [20,80,89,90,95,103,111] |
| 18 | Lack of collaboration and network establishment | Setting up a blockchain network system necessitates the participation and conviction of all involved parties that blockchain technology will offer them value. Moreover, achieving cooperation, communication, and coordination during the implementation process presents challenges. | [17,20,23,26,83,86,94,95,100,101,108,110,112,114,119,126,127] |
| 19 | Lack of maturity | Blockchain technology is currently in a developmental phase, and significant concerns persist regarding its resilience and robustness, particularly concerning large-scale transactions. This sensitivity surrounding blockchain adoption could stem from concerns about its reliability and resilience, particularly when handling substantial transactions. | [8,20,23,85,88,90,92,94,99–103,105,108,109,114,116,122,125,126] |
| 20 | Uncertain financial and environmental benefits | Blockchain mining consumes substantial energy for complex computations. Computers used for mining consume more energy than the rewards they offer. Expanding processor racks have energy requirements comparable to those necessary to light a megacity. Blockchain-based instruments' rising value attracts miners, yet increased rewards do not always lead to higher economic gains. Scalability issues worsen sustainability challenges. The development of proposed efficient hardware like application-specific integrated circuits lags in large-scale production. | [8,20,84,89,91–94,99,100,102,105,109,110,113] |
| 21 | Lack of employee acceptance | Employee acceptance hinges on multiple factors. Performance expectancy gauges whether the system aids job performance. Social influence considers others' views regarding system use. Facilitating conditions assess support for system use. In supply chains, transparency involves how information is conveyed to stakeholders. These factors shape employee system acceptance. | [8,19,78,85,98,101,107,119] |
| 22 | Reluctance to change business processes | Due to the broad applicability of blockchain technology across extensive and varied networks involving numerous stakeholders, be they individuals or institutions, its integration demands a specific level of expertise, time, and human resources. This could potentially lead companies and network participants to be hesitant to alter their business processes. | [8,89,91–95,105,109–111,113] |

**Table 1.** *Cont.*

| No. | Challenge | Description | References |
|---|---|---|---|
| 23 | Unclear responsibilities | While advocating for a decentralized database has its merits, there are instances where this kind of network can have drawbacks. Due to the distribution of data among participants in a blockchain, with equal footing depending on the permission type, the level of accountability among parties becomes ambiguous. | [91,105,109] |
| 24 | Lack of awareness | Customers' lack of awareness about blockchain technology and its diverse applications necessitates education regarding its features and implications for data ownership, access, and privacy. This education can boost blockchain adoption by companies and enhance the industry's social sustainability for customers. | [23,93,94,102,105,116,122] |

Quantitative data was required for the MCDM analysis utilized in this study to evaluate these challenges. An online questionnaire was deemed more suitable than alternative survey methods, such as focus groups, workshops, or interviews, due to its cost-effectiveness, automated data input and storage, broader coverage of the target group, reduced survey completion time, greater respondent availability, and consequently, higher response rates [128]. Between September and October 2023, the online questionnaire was distributed to selected respondents, comprising companies in the pharmaceutical industry, consulting firms, software companies, and academics. The emphasis was on engaging respondents with comprehensive and trans-disciplinary knowledge to avoid biases towards specific obstacles, excluding individuals with expertise solely in pharmaceuticals, sustainability, or blockchain technology.

Selection criteria mandated that experts possess a background in the pharmaceutical industry, expertise in sustainability, and practical knowledge of blockchain technology. Additionally, experts from various companies and departments were included to ensure access to a diverse range of information, thereby augmenting the reliability of the data. A total of 65 experts were invited to participate, with 18 consenting. The selected experts' background information is outlined in 21 challenges, as illustrated in Table 2.

**Table 2.** Experts' background information.

| No. | Gender | Highest Education | Industry | Expert Profile | Company | Address | Country | Total Professional Experience in Years |
|---|---|---|---|---|---|---|---|---|
| 1 | Male | MBA | IT | CIO | Digitalsoftlabs | Secunderabad | India | 23 |
| 2 | Male | Master | Pharma | Data Tech System Owner | Bayer | Leverkusen | Germany | 12 |
| 3 | Female | MBA | IT | Healthcare and Pharma Consultant | SAP | Zurich | Switzerland | 11 |
| 4 | Male | PhD | IT | Director | mSE | Chicago, IL | United States | 15 |
| 5 | Male | MBA | IT | Supply Chain Expert | SAP | London | UK | 25 |
| 6 | Male | Master | Pharma | Managing Director | ChemChain | Luxembourg | Luxembourg | 15 |
| 7 | Male | Master | Pharma | CEO | ARXUM France | Paris | France | 11 |
| 8 | Female | MBA | Pharma | Data Privacy Senior Associate | Bayer | Leverkusen | Germany | 11 |
| 9 | Male | Master | IT | Blockchain Scientist | SAP | Walldorf | Germany | 10 |
| 10 | Male | MBA | IT | Supply Chain Expert | SAP | Walldorf | Germany | 20 |

**Table 2.** *Cont.*

| No. | Gender | Highest Education | Industry | Expert Profile | Company | Address | Country | Total Professional Experience in Years |
|-----|--------|-------------------|----------|----------------|---------|---------|---------|----------------------------------------|
| 11 | Male | PhD | IT | Chief Product Manager Life Sciences | SAP | Walldorf | Germany | 30 |
| 12 | Male | MBA | IT | Managing Director | IT Services LargeCo | New Delhi | India | 27 |
| 13 | Female | Master | IT | Business Process Consultant | SAP | Toronto | Canada | 17 |
| 14 | Female | Professor | Academic | Professor | Polytechnic University of Milan | Milan | Italy | 30 |
| 15 | Female | MBA | IT | Managing Director | Ernst & Young | Irvine, CA | United States | 20 |
| 16 | Male | PhD | Pharma | Managing Director | PharmEng Technology | Toronto | Canada | 28 |
| 17 | Male | MBA | IT | Digital Supply Chain Manager | SAP SE | Istanbul | Turkey | 25 |
| 18 | Male | Professor | Academic | Associate Research Director | University of Cambridge | Singapore | Singapore | 16 |

While the literature still requires more consensus on the optimal number of experts in a judging panel, the expert group should ideally comprise 6 to 25 individuals [129]. A decision-making approach involving a small number of experts can be practical, if each expert possesses over ten years of experience [130]. The ISM method is particularly suitable for use with a few experts [131]. Moreover, previous studies employing the ISM methodology have reported the number of participating experts ranging from 5 to 15 [132–134]. To enhance the reliability of expert assessments, a series of online presentations was conducted to clarify the study's objectives.

Furthermore, experts were provided with a comprehensive guideline explaining each challenge. The final questionnaire underwent testing with a small sample of experts who contributed to the definitive list of barriers and provided insights into the final survey format. This ensured that the survey was concise, the definitions were comprehensive, and the questions were clear and easily comprehensible.

### 4.2. Data Analysis Using Statistical Tools

To identify a cluster of closely related challenges for more in-depth analysis, statistical techniques, specifically the importance index analysis and CIMTC, are applied. CIMTC quantifies the Pearson correlation coefficient between an individual item's score and the sum of scores for the remaining items. Items showing a weak correlation (CIMTC values below 0.3) with other items are excluded from further investigation [135].

Table 3 presents a comprehensive statistical overview, including CIMTC values, regarding the barriers to adopting blockchain technology in the pharmaceutical industry. It is evident that specific challenges, such as the "difficulty in changing organizational culture", "lack of maturity", and "uncertain financial environmental benefits", exhibit CIMTC values below 0.3. Consequently, they are not considered for analysis.

**Table 3.** Importance index analysis, challenges statistics, and CIMTC.

| No. | Challenge Code | Challenges of Blockchain Adoption in the Pharmaceutical Industry | Mean | Standard Deviation | Importance Index | CIMTC |
|---|---|---|---|---|---|---|
| 1 | $Ch_1$ | High costs of blockchain investments | 3.1111 | 1.2783 | 0.6222 | 0.5060 |
| 2 | $Ch_2$ | Lack of regulatory framework | 3.6667 | 1.0847 | 0.7333 | 0.6601 |
| 3 | $Ch_3$ | Lack of management support | 3.9444 | 1.0556 | 0.7889 | 0.5236 |
| 4 | $Ch_4$ | Lack of security | 2.8889 | 1.4507 | 0.5778 | 0.7194 |
| 5 | $Ch_5$ | Difficulty in changing organizational culture | 3.0556 | 1.3492 | 0.6111 | 0.2583 |
| 6 | $Ch_6$ | Lack of standardization and homogeneity | 3.1111 | 1.3235 | 0.7667 | 0.4913 |
| 7 | $Ch_7$ | Degree of immutability | 3.6667 | 1.4142 | 0.6222 | 0.7945 |
| 8 | $Ch_8$ | Lack of interoperability | 3.6111 | 1.0922 | 0.7444 | 0.6198 |
| 9 | $Ch_9$ | Lack of resources | 4.000 | 1.3720 | 0.7111 | 0.8437 |
| 10 | $Ch_{10}$ | Lack of governmental support and regulations | 3.3333 | 1.0290 | 0.7778 | 0.7332 |
| 11 | $Ch_{11}$ | Lack of market acceptance | 3.0000 | 1.2367 | 0.6778 | 0.7307 |
| 12 | $Ch_{12}$ | Lack of scalability | 3.2222 | 1.3086 | 0.5778 | 0.6181 |
| 13 | $Ch_{13}$ | Lack of maintenance and management | 3.3333 | 1.3284 | 0.6444 | 0.7692 |
| 14 | $Ch_{14}$ | Lack of technology expertise and skills | 3.1667 | 1.5811 | 0.7000 | 0.8541 |
| 15 | $Ch_{15}$ | Lack of data privacy | 3.1667 | 1.2005 | 0.5889 | 0.4668 |
| 16 | $Ch_{16}$ | Lack of validation | 2.8333 | 1.1504 | 0.6556 | 0.6219 |
| 17 | $Ch_{17}$ | Costs of latency | 3.1667 | 1.2485 | 0.5778 | 0.6527 |
| 18 | $Ch_{18}$ | Lack of collaboration and network establishment | 3.7222 | 1.3198 | 0.6667 | 0.6464 |
| 19 | $Ch_{19}$ | Lack of maturity | 3.9444 | 0.9984 | 0.7222 | 0.1380 |
| 20 | $Ch_{20}$ | Uncertain financial and environmental benefits | 2.7220 | 1.1785 | 0.7556 | 0.2942 |
| 21 | $Ch_{21}$ | Lack of employee acceptance | 3.1667 | 1.1504 | 0.5556 | 0.6152 |
| 22 | $Ch_{22}$ | Reluctance to change business processes | 3.1667 | 1.0432 | 0.6667 | 0.8038 |
| 23 | $Ch_{23}$ | Unclear responsibilities | 3.7222 | 0.9583 | 0.6333 | 0.4215 |
| 24 | $Ch_{24}$ | Lack of awareness | 3.7222 | 1.2744 | 0.7222 | 0.4126 |

The remaining 21 challenges display CIMTC values ranging from 0.4126 to 0.85410, making them suitable for further examination in this study. The survey data reveals a significant mean value, with a minimum of 2.8333 for all measures and a maximum standard deviation of 1.5811. This indicates that the data collected underscores the substantial significance of all the challenges identified for adopting blockchain technology in the pharmaceutical industry.

Additionally, the *importance index* analysis assesses the strength of expert opinions collected through the questionnaire survey. Numeric scores are transformed into relative importance indices using the formula provided in the equation below:

$$Importance\ index\ (I_x) = \frac{\sum_{i=1}^{5} p_i\, x_i}{5 \sum_{i=1}^{5} x_i} \tag{1}$$

In the equation, $p_i$ represents a constant that signifies the weight assigned to $i$, and $x_i$ represents a variable denoting the frequency of responses for $i$, which takes values from

1 to 5. The importance index spans from 0 to 1 and is categorized into five clusters to signify the respondent's rating, as depicted in the equation below:

$$
\begin{aligned}
&\text{Very important}: 0.8 < I_x \le 1.0 \\
&\text{Important}: 0.6 < I_x \le 0.8 \\
&\text{Preferred}: 0.4 < I_x \le 0.6 \\
&\text{Less important}: 0.2 < I_x \le 0.4 \\
&\text{Not important}: 0.0 < I_x \le 0.2
\end{aligned}
\tag{2}
$$

The analysis of the importance index for challenges in blockchain adoption was conducted using Equation (1), and the results are presented in Table 3. These results conclude that all barriers to blockchain adoption in the pharmaceutical industry are significant, as their importance index is above 0.2. A total of 16 barriers are classified as necessary, with an importance index exceeding 0.6 but less than 0.8. At the same time, the remaining five challenges are considered preferred, with an importance index above 0.4 but less than 0.6. The statistical analysis yields a roster of 21 established challenges. These confirmed challenges utilized in this study are listed in Table 4. Thus, RQ1 has been answered.

**Table 4.** Final list of challenges.

| No. | Challenge Code | Challenges of Blockchain Adoption in the Pharmaceutical Industry |
|:---:|:---:|:---:|
| 1 | $Ch_1$ | High costs of blockchain investments |
| 2 | $Ch_2$ | Lack of regulatory framework |
| 3 | $Ch_3$ | Lack of management support |
| 4 | $Ch_4$ | Lack of security |
| 5 | $Ch_5$ | Lack of standardization and homogeneity |
| 6 | $Ch_6$ | Degree of immutability |
| 7 | $Ch_7$ | Lack of interoperability |
| 8 | $Ch_8$ | Lack of resources |
| 9 | $Ch_9$ | Lack of governmental support and regulations |
| 10 | $Ch_{10}$ | Lack of market acceptance |
| 11 | $Ch_{11}$ | Lack of scalability |
| 12 | $Ch_{12}$ | Lack of maintenance and management |
| 13 | $Ch_{13}$ | Lack of technology expertise and skills |
| 14 | $Ch_{14}$ | Lack of data privacy |
| 15 | $Ch_{15}$ | Lack of validation |
| 16 | $Ch_{16}$ | Costs of latency |
| 17 | $Ch_{17}$ | Lack of collaboration and network establishment |
| 18 | $Ch_{18}$ | Lack of employee acceptance |
| 19 | $Ch_{19}$ | Reluctance to change business processes |
| 20 | $Ch_{20}$ | Unclear responsibilities |
| 21 | $Ch_{21}$ | Lack of awareness |

To mitigate bias, the consistency of the confirmed blockchain adoption challenges was evaluated through a reliability test conducted in SPSS software version 24. Data reliability was assessed by employing the Cronbach's alpha method. A Cronbach's alpha value closer to 1 indicates higher internal consistency reliability [136]. In this study, the alpha coefficient, with a value of 0.9211, exceeded 0.70, demonstrating its reliability and signifying strong internal consistency [137].

### 4.3. FISM Analysis

In the next phase, a comprehensive model utilizing the FISM approach has been constructed to ascertain the interconnected relationships among these factors within the Indian healthcare industry. The steps leading to the evolution of the model are given in 3.1.

#### 4.3.1. Creation of the Structural Self-Interaction Matrix

For this specific research, 18 experts from the pharmaceutical sector were invited to establish the interrelationships among variables influencing blockchain adoption in the pharmaceutical industry using a fuzzy scale ranging from 0 to 1. In this scale, 0 signifies no relationship, while 1 indicates a strong relationship among the variables. To represent the directional relationship between two challenges (i, j), four symbols are used:

- V is used if challenge "i" influences or reaches challenge "j";
- A is used if challenge "j" reaches challenge "i";
- X is used if challenge "i" and challenge "j" influence each other;
- O is used if both challenges are unrelated [138].

Considering the connections within the given context, the development of the SSIM has been undertaken for 21 challenges and is presented in Table 5.

**Table 5.** Structural self-interaction matrix.

| Challenge Code | $Ch_{21}$ | $Ch_{20}$ | $Ch_{19}$ | $Ch_{18}$ | $Ch_{17}$ | $Ch_{16}$ | $Ch_{15}$ | $Ch_{14}$ | $Ch_{13}$ | $Ch_{12}$ | $Ch_{11}$ | $Ch_{10}$ | $Ch_{9}$ | $Ch_{8}$ | $Ch_{7}$ | $Ch_{6}$ | $Ch_{5}$ | $Ch_{4}$ | $Ch_{3}$ | $Ch_{2}$ | $Ch_{1}$ |
|---|---|---|---|---|---|---|---|---|---|---|---|---|---|---|---|---|---|---|---|---|---|
| $Ch_{1}$ | V | O | V | O | V | V | V | V | V | A | V | V | O | A | V | O | V | V | V | V | X |
| $Ch_{2}$ | A | V | V | V | V | V | V | V | A | V | O | O | X | V | V | A | V | V | A | X | |
| $Ch_{3}$ | X | V | V | V | V | V | X | V | V | V | V | X | A | O | O | O | V | V | X | | |
| $Ch_{4}$ | A | A | A | X | V | V | V | X | A | A | O | V | A | A | O | A | A | X | | | |
| $Ch_{5}$ | O | O | O | O | O | O | A | O | O | V | V | O | A | O | X | O | X | | | | |
| $Ch_{6}$ | O | O | O | V | O | O | O | V | O | O | O | V | O | O | O | X | | | | | |
| $Ch_{7}$ | O | O | O | V | O | O | O | V | V | V | V | V | O | O | X | | | | | | |
| $Ch_{8}$ | A | O | A | V | O | O | V | V | X | V | V | V | O | X | | | | | | | |
| $Ch_{9}$ | V | V | V | V | V | V | V | X | V | V | V | X | X | | | | | | | | |
| $Ch_{10}$ | V | O | V | V | X | O | X | V | X | O | O | X | | | | | | | | | |
| $Ch_{11}$ | O | O | V | V | V | V | V | V | A | V | X | | | | | | | | | | |
| $Ch_{12}$ | O | A | V | X | V | V | V | V | A | X | | | | | | | | | | | |
| $Ch_{13}$ | V | V | V | X | V | V | X | V | X | | | | | | | | | | | | |
| $Ch_{14}$ | O | A | A | V | V | V | V | X | | | | | | | | | | | | | |
| $Ch_{15}$ | V | V | V | X | X | V | X | | | | | | | | | | | | | | |
| $Ch_{16}$ | V | A | V | V | V | X | | | | | | | | | | | | | | | |
| $Ch_{17}$ | X | A | V | X | X | | | | | | | | | | | | | | | | |
| $Ch_{18}$ | A | A | X | X | | | | | | | | | | | | | | | | | |
| $Ch_{19}$ | A | A | X | | | | | | | | | | | | | | | | | | |
| $Ch_{20}$ | A | X | | | | | | | | | | | | | | | | | | | |
| $Ch_{21}$ | X | | | | | | | | | | | | | | | | | | | | |

#### 4.3.2. Creation of the Reachability Matrix

After creating the SSIM, a reachability matrix is developed, which represents the accessibility of elements along a specific path [139]. The initial reachability matrix is obtained by altering each entry of the SSIM into 1s and 0s. The following rules are obeyed for the incorporation of binary entries:

- For the (i, j) entry, if it is A in SSIM, then the corresponding (i, j) entry in the reachability matrix becomes "1", and (j, i) becomes "0";
- For the (i, j) entry, if it is B in SSIM, then the corresponding (i, j) entry in the reachability matrix becomes "0", and (j, i) becomes "1";

- For the (i, j) entry, if it is C in SSIM, then the corresponding (i, j) entry in the reachability matrix becomes "1", and (j, i) becomes "1";
- for the (i, j) entry, if it is D in SSIM, then the corresponding (i, j) entry in the reachability matrix becomes "0", and (j, i) becomes "0" [140].

The initial reachability matrix is then checked for transitivity, which refers to a relationship involving three variables, wherein if a connection is identified between the first and second variables, as well as between the second and third variables, it logically implies the existence of a relationship between the first and third variables. By incorporating 1* in the initial reachability matrix to address any potential judgmental gaps that may arise following the collection of experts' opinions, transitivity is assumed, and the final reachability matrix is developed, as represented in Table 6.

**Table 6.** Final reachability matrix.

| Challenge Code | $Ch_1$ | $Ch_2$ | $Ch_3$ | $Ch_4$ | $Ch_5$ | $Ch_6$ | $Ch_7$ | $Ch_8$ | $Ch_9$ | $Ch_{10}$ | $Ch_{11}$ | $Ch_{12}$ | $Ch_{13}$ | $Ch_{14}$ | $Ch_{15}$ | $Ch_{16}$ | $Ch_{17}$ | $Ch_{18}$ | $Ch_{19}$ | $Ch_{20}$ | $Ch_{21}$ |
|---|---|---|---|---|---|---|---|---|---|---|---|---|---|---|---|---|---|---|---|---|---|
| $Ch_1$ | 1 | 1 | 1 | 1 | 1 | 1 * | 1 | 0 | 0 | 1 | 1 | 0 | 1 | 1 | 1 | 1 | 1 | 1 * | 1 | 1 * | 1 |
| $Ch_2$ | 0 | 1 | 0 | 1 | 1 | 0 | 1 | 1 | 1 | 0 | 0 | 1 | 0 | 1 | 1 | 1 | 1 | 1 | 1 | 1 | 0 |
| $Ch_3$ | 0 | 1 | 1 | 1 | 1 | 0 | 1 * | 1 * | 0 | 1 | 1 | 1 | 1 | 1 | 1 | 1 | 1 | 1 | 1 | 1 | 1 |
| $Ch_4$ | 0 | 0 | 0 | 1 | 0 | 0 | 0 | 0 | 0 | 1 | 0 | 0 | 0 | 1 | 1 | 1 | 1 | 1 | 0 | 0 | 0 |
| $Ch_5$ | 0 | 0 | 1 | 1 | 1 | 0 | 1 | 1 * | 0 | 1 * | 0 | 0 | 0 | 1 * | 0 | 1 * | 1 * | 1 * | 0 | 0 | 0 |
| $Ch_6$ | 0 | 1 | 0 | 1 | 1 * | 1 | 1 * | 1 * | 1 * | 1 | 1 * | 1 * | 1 * | 1 | 1 * | 1 * | 1 * | 1 | 1 * | 1 * | 1 * |
| $Ch_7$ | 0 | 0 | 0 | 1 * | 1 | 0 | 1 | 1 * | 1 * | 1 | 1 | 1 | 1 | 1 | 1 * | 1 * | 1 * | 1 | 1 * | 1 * | 1 * |
| $Ch_8$ | 1 | 0 | 1 * | 1 | 1 * | 0 | 1 * | 1 | 1 * | 1 | 1 | 1 | 1 | 1 | 1 | 1 * | 1 * | 1 | 0 | 1 * | 0 |
| $Ch_9$ | 1 * | 1 | 1 | 1 | 1 | 0 | 1 * | 1 * | 1 | 1 | 1 | 1 | 1 | 1 | 1 | 1 | 1 | 1 | 1 | 1 | 1 |
| $Ch_{10}$ | 0 | 1 * | 1 | 0 | 1 * | 0 | 0 | 0 | 1 | 1 | 1 * | 1 * | 1 | 1 | 1 | 1 * | 1 | 1 | 1 | 1 * | 1 |
| $Ch_{11}$ | 0 | 1 | 0 | 0 | 0 | 0 | 0 | 0 | 0 | 0 | 1 | 1 | 1 * | 1 | 1 | 1 | 1 | 1 | 1 | 1 * | 1 * |
| $Ch_{12}$ | 1 | 0 | 0 | 1 | 0 | 0 | 0 | 0 | 0 | 1 * | 0 | 1 | 0 | 1 | 1 | 1 | 1 | 1 | 1 | 0 | 1 * |
| $Ch_{13}$ | 0 | 1 | 0 | 1 | 1 * | 0 | 0 | 1 | 0 | 1 | 1 | 1 | 1 | 1 | 1 | 1 | 1 | 1 | 1 | 1 | 1 |
| $Ch_{14}$ | 0 | 0 | 0 | 1 | 0 | 0 | 0 | 0 | 1 | 0 | 0 | 0 | 0 | 1 | 1 | 1 | 1 | 1 | 0 | 0 | 1 * |
| $Ch_{15}$ | 0 | 0 | 1 | 0 | 1 | 0 | 0 | 0 | 0 | 1 | 0 | 0 | 1 | 0 | 1 | 1 | 1 | 1 | 1 | 1 | 1 |
| $Ch_{16}$ | 0 | 0 | 0 | 0 | 0 | 0 | 0 | 1 * | 0 | 1 * | 0 | 0 | 0 | 0 | 0 | 1 | 1 | 1 | 1 | 0 | 1 |
| $Ch_{17}$ | 0 | 0 | 0 | 0 | 0 | 0 | 0 | 1 * | 0 | 1 | 0 | 0 | 0 | 0 | 1 | 0 | 1 | 1 | 1 | 0 | 1 |
| $Ch_{18}$ | 0 | 0 | 0 | 1 | 1 * | 0 | 0 | 0 | 0 | 0 | 0 | 1 | 1 | 0 | 1 | 0 | 1 | 1 | 1 | 0 | 0 |
| $Ch_{19}$ | 0 | 0 | 0 | 1 | 0 | 0 | 0 | 1 | 0 | 0 | 0 | 0 | 0 | 1 | 0 | 0 | 0 | 1 | 1 | 0 | 0 |
| $Ch_{20}$ | 1 * | 0 | 0 | 1 | 0 | 0 | 0 | 1 * | 0 | 1 * | 1 * | 1 | 0 | 1 | 1 | 1 | 1 | 1 | 1 | 1 | 0 |
| $Ch_{21}$ | 0 | 1 | 1 | 1 | 0 | 0 | 1 * | 1 | 0 | 0 | 1 * | 1 * | 0 | 1 * | 0 | 0 | 1 | 1 | 1 | 1 | 1 |

* represents transitivity.

### 4.3.3. Partitioning of the Final Reachability Matrix into Different Levels

The final reachability matrix derived in Section 4.3.2 was segmented into distinct levels. This facilitated determining the reachability and antecedent sets for each barrier, according to Ref. [141]. The reachability set for the finalized challenges encompasses the challenges themselves and other achievement enablers that they may contribute. In the row corresponding to a specific considered factor, each column containing a 1 is included in the reachability set, representing the factor associated with that column. Conversely, the antecedent set comprises the challenges and other complications that may give rise to them. The intersection of these sets was also calculated for all challenges. In the column corresponding to the considered factor, the antecedent set includes the factors represented by rows containing a value of 1. If the reachability set and the intersection set for a given barrier are identical, this situation is categorized as Level I and occupies the lowest position in the ISM hierarchy. This process marks the completion of iteration 1. Subsequently, the barriers identified in Level I are discarded, and the procedure continues with the remaining barriers in iteration 2. This iterative approach persists until the level of each barrier is determined [142]. The compiled iterations of the challenges are outlined in Table 7.

**Table 7.** Levels of challenges.

| Ch (Ch$_i$) | Reachability Set | Antecedent Set | Intersection | Level |
|---|---|---|---|---|
| Ch$_1$ | 1, 2, 3, 4, 5, 7, 10, 11, 13, 14, 15, 16, 17, 18, 19, 20, 21 | 1, 8, 9, 12, 18, 20 | 1, 18, 20 | V |
| Ch$_2$ | 2, 4, 5, 7, 8, 9, 12, 14, 15, 16, 17, 18, 19, 20 | 1, 2, 3, 6, 9, 10, 13, 21 | 2, 9 | IV |
| Ch$_3$ | 2, 3, 4, 5, 7, 8, 10, 11, 12, 13, 14, 15, 16, 17, 18, 19, 20, 21 | 1, 3, 6, 8, 9, 10, 15, 21 | 3, 8, 10, 15, 21 | V |
| Ch$_4$ | 4, 10, 14, 15, 16, 17, 18 | 1, 2, 3, 4, 5, 6, 7, 8, 9, 12, 13, 14, 18, 19, 20, 21 | 4, 14, 18 | III |
| Ch$_5$ | 4, 5, 7, 8, 10, 12, 13, 14, 16, 17, 18 | 1, 2, 3, 5, 6, 7, 8, 9, 10, 13, 14, 15, 16, 17, 18 | 5, 7, 8, 10, 13, 14, 16, 17, 18 | II |
| Ch$_6$ | 2, 3, 4, 5, 6, 7, 8, 9, 10, 11, 12, 13, 14, 15, 16, 17, 18, 19, 20, 21 | 6, 11, 12, 13, 15, 16, 17, 19, 20, 21 | 6, 11, 12, 13, 15, 16, 17, 19, 20, 21 | V |
| Ch$_7$ | 4, 5, 7, 8, 9, 10, 11, 12, 13, 14, 15, 16, 17, 18, 19, 20, 21 | 1, 2, 3, 5, 6, 7, 8, 9, 15, 16, 17, 19, 20, 21 | 5, 7, 8, 9, 15, 16, 17, 19, 20, 21 | IV |
| Ch$_8$ | 1, 3, 4, 5, 7, 8, 9, 10, 11, 12, 13, 14, 15, 16, 17, 18, 20 | 2, 3, 5, 6, 7, 8, 9, 13, 16, 17, 19, 20, 21 | 3, 5, 7, 8, 9, 13, 16, 17, 20 | IV |
| Ch$_9$ | 1, 2, 3, 4, 5, 7, 8, 9, 10, 11, 12, 13, 14, 15, 16, 17, 18, 19, 20, 21 | 2, 6, 7, 8, 9, 10, 14 | 2, 7, 8, 9, 10, 14 | IV |
| Ch$_{10}$ | 2, 3, 5, 9, 10, 11, 12, 13, 14, 15, 16, 17, 18, 19, 20, 21 | 1, 3, 4, 5, 6, 7, 8, 9, 10, 11, 12, 13, 15, 16, 17, 20 | 3, 5, 9, 10, 11, 12, 13, 15, 16, 17, 20 | V |
| Ch$_{11}$ | 6, 10, 11, 12, 14, 15, 16, 17, 18, 20, 21 | 1, 3, 6, 7, 8, 9, 10, 11, 13, 19, 20, 21 | 6, 10, 11, 20, 21 | III |
| Ch$_{12}$ | 1, 4, 6, 10, 12, 14, 15, 16, 17, 18, 21 | 2, 3, 5, 6, 7, 8, 9, 10, 11, 12, 13, 18, 19, 20, 21 | 6, 10, 12, 18, 21 | IV |
| Ch$_{13}$ | 2, 4, 5, 6, 8, 10, 11, 12, 13, 14, 15, 16, 17, 18, 19, 20, 21 | 1, 3, 5, 6, 7, 8, 9, 10, 13, 15, 18 | 5, 6, 8, 10, 13, 15, 18 | V |
| Ch$_{14}$ | 4, 5, 9, 14, 15, 16, 17, 18, 21 | 1, 2, 3, 4, 5, 6, 7, 8, 9, 10, 11, 12, 13, 14, 19, 20, 21 | 4, 5, 9, 14, 21 | III |
| Ch$_{15}$ | 3, 5, 6, 7, 10, 13, 15, 16, 17, 18, 19, 20, 21 | 1, 2, 3, 4, 6, 7, 8, 9, 10, 11, 12, 13, 14, 15, 17, 18, 20 | 3, 5, 6, 7, 10, 13, 15, 17, 18 | III |
| Ch$_{16}$ | 5, 6, 7, 8, 10, 16, 17, 18, 19, 21 | 1, 2, 3, 4, 5, 6, 7, 8, 9, 10, 11, 12, 13, 14, 15, 16, 20 | 5, 6, 7, 8, 10, 16 | II |
| Ch$_{17}$ | 5, 6, 7, 8, 10, 15, 17, 18, 19, 21 | 1, 2, 3, 4, 5, 6, 7, 8, 9, 10, 11, 12, 13, 14, 15, 16, 17, 18, 19, 20, 21 | 5, 6, 7, 8, 10, 15, 17, 18, 19, 21 | I |
| Ch$_{18}$ | 1, 4, 5, 12, 13, 15, 17, 18, 19 | 1, 2, 3, 4, 5, 6, 7, 8, 9, 10, 11, 12, 13, 14, 15, 16, 17, 18, 19, 20, 21 | 1, 4, 5, 12, 13, 15, 17, 18, 19 | I |
| Ch$_{19}$ | 4, 6, 7, 8, 11, 12, 14, 17, 18 19 | 1, 2, 3, 6, 7, 9, 10, 13, 15, 16, 17, 18, 19, 20, 21 | 6, 7, 18, 19 | IV |
| Ch$_{20}$ | 1, 4, 6, 7, 8, 10, 11, 12, 14, 15, 16, 17, 18, 19, 20 | 1, 2, 3, 6, 7, 8, 9, 10, 11, 13, 15, 20, 21 | 1, 6, 7, 8, 10, 11, 20 | III |
| Ch$_{21}$ | 2, 3, 4, 6, 7, 8, 11, 12, 14, 17, 18, 19, 20, 21 | 1, 3, 6, 7, 9, 10, 11, 12, 13, 14, 15, 16, 17, 21 | 3, 6, 7, 11, 12, 14, 17, 21 | V |

### 4.3.4. Development of the Fuzzy Conical Matrix

FISM represents an advancement over traditional ISM methodology. In FISM, an additional input, the possibility of interaction, is introduced on a 0–1 scale, excluding both 0 and 1, so the traditional binary representation of relationships as 0 and 1 is replaced with quantifiable values on a fuzzy scale, providing a more nuanced data representation [143]. Unlike ISM, FISM incorporates a fuzzy scale. The fuzzy scale utilized in this study is detailed in Table 8 [144].

**Table 8.** Possibility of numerical value of the reachability.

| Possibility of Reachability | No. | Very Low | Low | Medium | High | Very High |
|---|---|---|---|---|---|---|
| Value | 0 | 0.1 | 0.3 | 0.5 | 0.7 | 0.9 |

The final reachability matrix is then transformed based on this chosen fuzzy scale, resulting in a fuzzy final reachability matrix. This fuzzy reachability matrix is subsequently utilized to determine the dependence and driving power of the variables, as detailed in

Table 9. A fuzzy conical matrix is constructed to streamline the analysis by consolidating factors at the same level across different rows and columns of the fuzzy final reachability matrix, as illustrated in Table 10. The drive power of a factor is computed by summing the number of ones in its corresponding rows, while the dependence power is calculated by summing the number of ones in its columns. To determine the rankings for dependence and drive power, the highest ranks are assigned to the factors with the most significant number of "ones" in their respective rows and columns.

**Table 9.** Fuzzy final reachability matrix.

| Challenge Code | Ch$_1$ | Ch$_2$ | Ch$_3$ | Ch$_4$ | Ch$_5$ | Ch$_6$ | Ch$_7$ | Ch$_8$ | Ch$_9$ | Ch$_{10}$ | Ch$_{11}$ | Ch$_{12}$ | Ch$_{13}$ | Ch$_{14}$ | Ch$_{15}$ | Ch$_{16}$ | Ch$_{17}$ | Ch$_{18}$ | Ch$_{19}$ | Ch$_{20}$ | Ch$_{21}$ |
|---|---|---|---|---|---|---|---|---|---|---|---|---|---|---|---|---|---|---|---|---|---|
| Ch$_1$ | 1 | 1 | 1 | 1 | 1 | 0.1 | 1 | 0 | 0 | 1 | 1 | 0 | 1 | 1 | 1 | 1 | 1 | 0.3 | 1 | 0.1 | 1 |
| Ch$_2$ | 0 | 1 | 0 | 1 | 1 | 0 | 1 | 1 | 1 | 0 | 0 | 1 | 0 | 1 | 1 | 1 | 1 | 1 | 1 | 1 | 0 |
| Ch$_3$ | 0 | 1 | 1 | 1 | 1 | 0 | 0.5 | 0.5 | 0 | 1 | 1 | 1 | 1 | 1 | 1 | 1 | 1 | 1 | 1 | 1 | 1 |
| Ch$_4$ | 0 | 0 | 0 | 1 | 0 | 0 | 0 | 0 | 0 | 1 | 0 | 0 | 0 | 1 | 1 | 1 | 1 | 1 | 0 | 0 | 0 |
| Ch$_5$ | 0 | 0 | 1 | 1 | 1 | 0 | 1 | 0.1 | 0 | 0.5 | 0 | 0 | 0 | 0.3 | 0 | 0.1 | 0.3 | 0.3 | 0 | 0 | 0 |
| Ch$_6$ | 0 | 1 | 0 | 1 | 0.1 | 1 | 0.1 | 0.1 | 0.1 | 1 | 0.1 | 0.3 | 0.5 | 1 | 0.1 | 0.1 | 0.3 | 1 | 0.1 | 0.1 | 0.3 |
| Ch$_7$ | 0 | 0 | 0 | 0.3 | 1 | 0 | 1 | 0.3 | 0.1 | 1 | 1 | 1 | 1 | 1 | 0.5 | 0.1 | 0.5 | 1 | 0.1 | 0.1 | 0.3 |
| Ch$_8$ | 1 | 0 | 0.5 | 1 | 0.3 | 0 | 0.1 | 1 | 0.1 | 1 | 1 | 1 | 1 | 1 | 1 | 0.1 | 0.5 | 1 | 0 | 0.1 | 0 |
| Ch$_9$ | 0.3 | 1 | 1 | 1 | 1 | 0 | 0.1 | 0.5 | 1 | 1 | 1 | 1 | 1 | 1 | 1 | 1 | 1 | 1 | 1 | 1 | 1 |
| Ch$_{10}$ | 0 | 0.3 | 1 | 0 | 0.5 | 0 | 0 | 0 | 1 | 1 | 0.5 | 0.3 | 1 | 1 | 1 | 0.3 | 1 | 1 | 1 | 0.1 | 1 |
| Ch$_{11}$ | 0 | 1 | 0 | 0 | 0 | 0 | 0 | 0 | 0 | 0 | 1 | 1 | 0.3 | 1 | 1 | 1 | 1 | 1 | 1 | 0.1 | 0.3 |
| Ch$_{12}$ | 1 | 0 | 0 | 1 | 0 | 0 | 0 | 0 | 0 | 0.1 | 0 | 1 | 0 | 1 | 1 | 1 | 1 | 1 | 1 | 0 | 0.3 |
| Ch$_{13}$ | 0 | 1 | 0 | 1 | 0.5 | 0 | 0 | 1 | 0 | 1 | 1 | 1 | 1 | 1 | 1 | 1 | 1 | 1 | 1 | 1 | 1 |
| Ch$_{14}$ | 0 | 0 | 0 | 1 | 0 | 0 | 0 | 0 | 1 | 0 | 0 | 0 | 0 | 1 | 1 | 1 | 1 | 1 | 0 | 0 | 0.5 |
| Ch$_{15}$ | 0 | 0 | 1 | 0 | 1 | 0 | 0 | 0 | 0 | 1 | 0 | 0 | 1 | 0 | 1 | 1 | 1 | 1 | 1 | 1 | 1 |
| Ch$_{16}$ | 0 | 0 | 0 | 0 | 0 | 0 | 0 | 0.1 | 0 | 0.3 | 0 | 0 | 0 | 0 | 0 | 1 | 1 | 1 | 1 | 0 | 1 |
| Ch$_{17}$ | 0 | 0 | 0 | 0 | 0 | 0 | 0 | 0.7 | 0 | 1 | 0 | 0 | 0 | 0 | 1 | 0 | 1 | 1 | 1 | 0 | 1 |
| Ch$_{18}$ | 0 | 0 | 0 | 1 | 0.5 | 0 | 0 | 0 | 0 | 0 | 0 | 1 | 1 | 0 | 1 | 0 | 1 | 1 | 1 | 0 | 0 |
| Ch$_{19}$ | 0 | 0 | 0 | 1 | 0 | 0 | 0 | 1 | 0 | 0 | 0 | 0 | 0 | 1 | 0 | 0 | 0 | 1 | 1 | 0 | 0 |
| Ch$_{20}$ | 0.1 | 0 | 0 | 1 | 0 | 0 | 0 | 0.1 | 0 | 0.1 | 0.1 | 1 | 0 | 1 | 1 | 1 | 1 | 1 | 1 | 1 | 0 |
| Ch$_{21}$ | 0 | 1 | 1 | 1 | 0 | 0 | 0.3 | 1 | 0 | 0 | 0.5 | 0.3 | 0 | 0.5 | 0 | 0 | 1 | 1 | 1 | 1 | 1 |

**Table 10.** Fuzzy conical matrix.

| Challenge Code | Ch$_{19}$ | Ch$_{16}$ | Ch$_4$ | Ch$_5$ | Ch$_{17}$ | Ch$_{14}$ | Ch$_{18}$ | Ch$_6$ | Ch$_{20}$ | Ch$_{12}$ | Ch$_{11}$ | Ch$_7$ | Ch$_{21}$ | Ch$_{15}$ | Ch$_8$ | Ch$_{10}$ | Ch$_2$ | Ch$_1$ | Ch$_{13}$ | Ch$_3$ | Ch$_9$ | Driving Power |
|---|---|---|---|---|---|---|---|---|---|---|---|---|---|---|---|---|---|---|---|---|---|---|
| Ch$_{19}$ | 1 | 0 | 1 | 0 | 0 | 1 | 1 | 0 | 0 | 0 | 0 | 0 | 0 | 0 | 1 | 0 | 0 | 0 | 0 | 0 | 0 | 5 |
| Ch$_{16}$ | 1 | 1 | 0 | 0 | 1 | 0 | 1 | 0 | 0 | 0 | 0 | 0 | 0 | 1 | 0 | 0.1 | 0.3 | 0 | 0 | 0 | 0 | 5.4 |
| Ch$_4$ | 1 | 1 | 1 | 0 | 1 | 1 | 1 | 0 | 0 | 0 | 0 | 0 | 0 | 1 | 0 | 1 | 0 | 0 | 0 | 0 | 0 | 8 |
| Ch$_5$ | 0 | 0.1 | 1 | 1 | 0.3 | 0.3 | 0.3 | 0 | 0 | 0 | 0 | 1 | 0 | 0 | 0.1 | 0.5 | 0 | 0 | 0 | 1 | 0 | 5.6 |
| Ch$_{17}$ | 0 | 0 | 0 | 0 | 1 | 0 | 1 | 0 | 0 | 0 | 0 | 0 | 1 | 1 | 0.7 | 1 | 0 | 0 | 0 | 0 | 0 | 6.7 |
| Ch$_{14}$ | 0.1 | 1 | 1 | 0 | 1 | 1 | 1 | 0 | 0 | 0 | 0 | 0 | 0 | 0.5 | 1 | 0 | 0 | 0 | 0 | 0 | 1 | 7.5 |
| Ch$_{18}$ | 0.1 | 0 | 1 | 0.5 | 1 | 0 | 1 | 0 | 0 | 1 | 0 | 0 | 0 | 1 | 0 | 0 | 0 | 0 | 1 | 0 | 0 | 7.5 |
| Ch$_6$ | 0 | 0.1 | 1 | 0.1 | 0.3 | 1 | 1 | 1 | 0.1 | 0.3 | 0.1 | 0.1 | 0.3 | 0.1 | 0.1 | 1 | 1 | 0 | 0.5 | 0 | 0.1 | 8.3 |
| Ch$_{20}$ | 1 | 1 | 1 | 0 | 1 | 1 | 1 | 0 | 1 | 1 | 0.1 | 0 | 0 | 0 | 0.3 | 0.1 | 0 | 0.1 | 0 | 0 | 0 | 8.6 |
| Ch$_{12}$ | 1 | 1 | 1 | 0 | 1 | 1 | 1 | 0 | 0 | 1 | 0 | 0 | 0.3 | 1 | 0 | 0.1 | 0 | 1 | 0 | 0 | 0 | 9.4 |
| Ch$_{11}$ | 1 | 1 | 0 | 0 | 1 | 1 | 1 | 0 | 0.1 | 1 | 1 | 0 | 0.3 | 1 | 0 | 0 | 1 | 0 | 0.3 | 0 | 0 | 9.7 |
| Ch$_7$ | 1 | 0.1 | 0.3 | 1 | 0.5 | 1 | 1 | 0 | 0.1 | 1 | 1 | 1 | 0.3 | 0.5 | 0.3 | 1 | 0 | 0 | 1 | 0 | 0.1 | 10.3 |
| Ch$_{21}$ | 1 | 0 | 1 | 0 | 1 | 0.5 | 1 | 0 | 1 | 0.3 | 0.5 | 0.3 | 1 | 0 | 1 | 0 | 1 | 0 | 0 | 1 | 0 | 10.7 |
| Ch$_{15}$ | 0 | 1 | 0 | 1 | 1 | 0 | 1 | 0 | 1 | 0 | 0 | 0 | 0 | 1 | 1 | 0 | 1 | 0 | 0 | 1 | 1 | 10 |
| Ch$_8$ | 1 | 0.1 | 1 | 0.3 | 0.5 | 1 | 1 | 0 | 0.1 | 1 | 1 | 0.1 | 0 | 1 | 1 | 1 | 0 | 1 | 1 | 0.5 | 0.1 | 11.7 |
| Ch$_{10}$ | 1 | 0.3 | 0 | 0.5 | 1 | 1 | 1 | 0 | 0.1 | 0.3 | 0.5 | 0 | 1 | 1 | 0 | 1 | 0.3 | 0 | 1 | 1 | 1 | 12 |
| Ch$_2$ | 1 | 1 | 1 | 1 | 1 | 1 | 1 | 0 | 1 | 1 | 0 | 1 | 0 | 1 | 1 | 0 | 1 | 0 | 0 | 0 | 1 | 14 |
| Ch$_1$ | 1 | 1 | 1 | 1 | 1 | 1 | 0.3 | 0.1 | 0.1 | 0 | 1 | 1 | 1 | 1 | 0 | 1 | 1 | 1 | 1 | 1 | 0 | 15.5 |
| Ch$_{13}$ | 1 | 1 | 1 | 0.5 | 1 | 1 | 1 | 0 | 1 | 1 | 1 | 0 | 1 | 1 | 1 | 1 | 1 | 0 | 1 | 0 | 0 | 15.5 |

**Table 10.** *Cont.*

| Challenge Code | $Ch_{19}$ | $Ch_{16}$ | $Ch_4$ | $Ch_5$ | $Ch_{17}$ | $Ch_{14}$ | $Ch_{18}$ | $Ch_6$ | $Ch_{20}$ | $Ch_{12}$ | $Ch_{11}$ | $Ch_7$ | $Ch_{21}$ | $Ch_{15}$ | $Ch_8$ | $Ch_{10}$ | $Ch_2$ | $Ch_1$ | $Ch_{13}$ | $Ch_3$ | $Ch_9$ | Driving Power |
|---|---|---|---|---|---|---|---|---|---|---|---|---|---|---|---|---|---|---|---|---|---|---|
| $Ch_3$ | 1 | 1 | 1 | 1 | 1 | 1 | 1 | 0 | 1 | 1 | 1 | 0.5 | 1 | 1 | 0.5 | 1 | 1 | 0 | 1 | 1 | 0 | 17 |
| $Ch_9$ | 1 | 1 | 1 | 1 | 1 | 1 | 1 | 0 | 1 | 1 | 1 | 0.1 | 1 | 1 | 0.5 | 1 | 1 | 0.3 | 1 | 1 | 1 | 17.9 |
| Dependence power | 15.2 | 12.6 | 15.3 | 8.9 | 17.6 | 15.8 | 19.6 | 1.1 | 7.6 | 10.9 | 8.2 | 5.1 | 10.7 | 14.6 | 7.6 | 12 | 8.3 | 3.4 | 9.8 | 7.5 | 4.3 | 215.2/215.2 |

### 4.3.5. Creation of the FISM Model Fuzzy Conical Matrix

Considering the fuzzy conical matrix, an FISM model is developed based on the relationships given above in the fuzzy final reachability matrix. The structural model is derived from the final reachability matrix and the segmented levels. If a relationship exists between challenges i and j, it is represented by an arrow pointing from i to j in a directed graph or diagraph. This diagraph illustrates all potential dependencies and transitivities among the challenges generated from one level to another. The concept of transitivity in challenges is acknowledged, meaning that if challenges i are related to j and challenges j are related to k, then challenges i would also be considered related to k [145]. Ultimately, the nodes of all elements are replaced with corresponding statements, and the digraph is transformed into an FISM model, as shown in Figure 3.

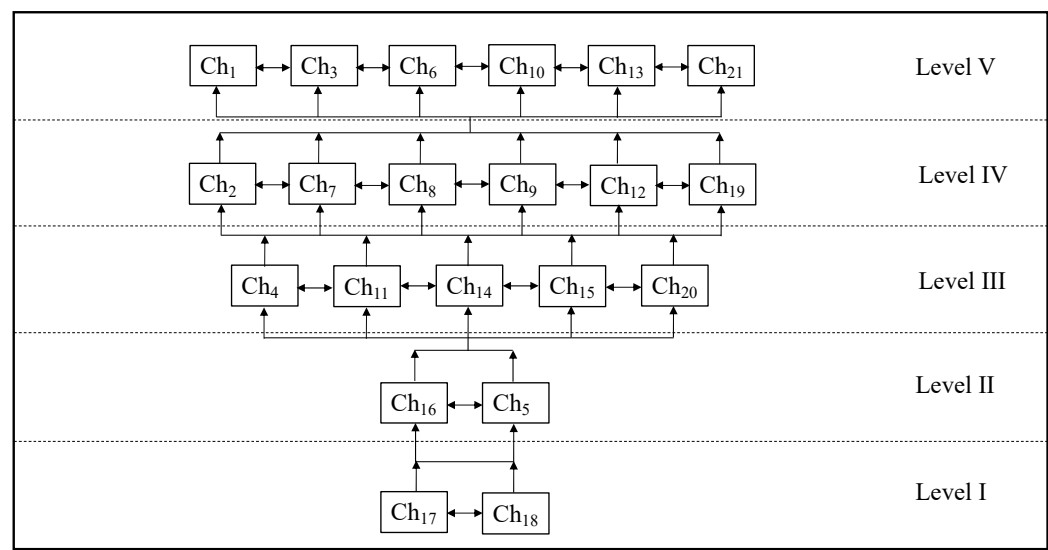

**Figure 3.** FISM model.

### 4.4. MICMAC Analysis

The MICMAC analysis is conducted based on the matrix multiplication principle, aiming to assess each construct's driving and dependence force and to subsequently categorize the research variables accordingly [68]. The dependence and driving power of the factors are illustrated in Table 10. Subsequently, a diagram depicting the dependence and driving power is generated in Figure 4, including clusters of challenges affecting blockchain adoption. There are three challenges exhibiting weak driving power and weak dependency, which are unrelated to the other barriers in the model. Seven challenges are included in the second cluster (weak driving power, strong dependency); thus, these challenges occupy elevated positions in the ISM-based model. Strong driving power and high dependency can be found in four challenges, which means that they play a crucial role in interconnectedness and dynamics. Independent challenges show strong driving power, as well as a substantial dependency. The seven challenges occurring in the fourth cluster are strategically important, and changes in these factors can have significant implications for the overall behavior of the challenge. By displaying the existing connections among the challenges related to blockchain adoption in the pharmaceutical industry, RQ2 has been addressed.

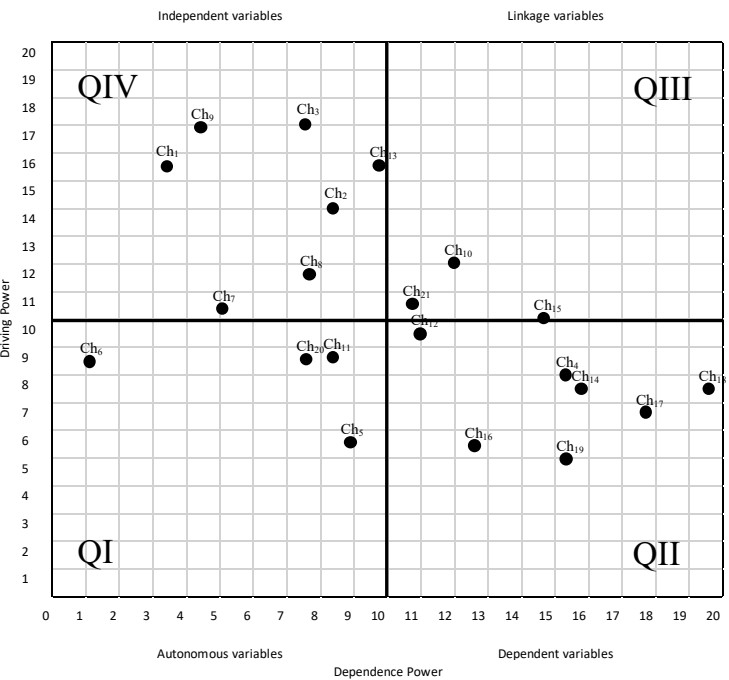

**Figure 4.** Clusters of challenges affecting blockchain adoption.

## 5. Discussion and Implications

This research aims to assess the significant factors influencing the adoption of blockchain technology in the pharmaceutical industry. The objective is to provide pharmaceutical management and policymakers with insights for more effective and efficient handling of these factors. Utilizing the FISM approach, a comprehensive model has been developed to thoroughly examine the interrelationships among these factors.

### 5.1. Discussion

Although the implementation of blockchain technology shows great potential for fostering sustainable development in the pharmaceutical industry, and despite an increasing amount of research regarding the hurdles linked to its adoption, a thorough comprehension of the potential challenges regarding embracing blockchain is still in its infancy. Therefore, this study empirically analyzes blockchain adoption challenges in the pharmaceutical industry. This model assists in establishing a hierarchy of actions and activities which can be undertaken by management to address the noteworthy impacts of blockchain adoption within the pharmaceutical industry. This information is crucial for decision makers and policymakers to formulate effective strategies promoting blockchain adoption in the pharmaceutical sector.

The findings present five distinct dimensions describing relationships among the selected barrier factors. The first level in the hierarchy encompasses $Ch_{17}$ and $Ch_{18}$. The second level includes $Ch_5$ and $Ch_{16}$. The third level involves $Ch_4$, $Ch_{11}$, $Ch_{14}$, $Ch_{15}$, and $Ch_{20}$. The fourth level comprises $Ch_2$, $Ch_7$, $Ch_8$, $Ch_9$, $Ch_{12}$, and $Ch_{19}$. The fifth level highlights $Ch_1$, $Ch_3$, $Ch_6$, $Ch_{10}$, $Ch_{13}$, and $Ch_{21}$. Figure 3 illustrates that $Ch_1$, $Ch_3$, $Ch_6$, $Ch_{10}$, $Ch_{13}$, and $Ch_{21}$ represent the most significant challenges to blockchain adoption in the pharmaceutical industry. Positioned at the bottom of the entire structural hierarchy, they form the foundation of the entire model. Some of these observations can be explained by the distinctive features of the current pharmaceutical industry, whereas others are generic. The sector, known for its complex supply chains and stringent regulatory requirements, places a premium on secure data management.

However, implementing blockchain solutions, which demand significant investments in technology infrastructure and maintenance, becomes a formidable financial challenge. Justifying these costs is further complicated by the industry's already substantial research,

development, and compliance expenditures. This economic challenge is closely linked to the need for solid leadership support for successful blockchain adoption. Without total management commitment, justifying high costs becomes more complex and hampers resource allocation, decision-making processes, and the overall integration of blockchain technology into existing systems. Widespread unfamiliarity with blockchain technology and its potential applications is not only an issue fostering the lack of management support but instead poses a significant general challenge. Many stakeholders, ranging from industry professionals to regulators and the public, may need to fully grasp blockchain's intricacies and advantages. The imperative lies in conducting extensive educational initiatives to raise awareness and demystify blockchain technology, ensuring a more informed and receptive audience. Industry associations, regulatory bodies, and influential stakeholders can play a pivotal role in disseminating information. Implementing educational programs targeting professionals at all levels will contribute to a broader understanding of blockchain technology's potential and benefits. The need for more professionals with specialized blockchain skills exacerbates these challenges.

The pharmaceutical industry requires help finding and retaining talent proficient in cryptography, distributed ledger technology, and smart contract development. This shortage slows down the adoption process and jeopardizes the effective utilization of blockchain technology within the industry. The emphasis on data accuracy and integrity in the pharmaceutical industry is intricately tied to the immutability of blockchain technology. While blockchain technology ensures data integrity, the challenge arises in situations requiring flexibility, such as regulatory changes or corrections to erroneous data, where the inability to alter information becomes a delicate consideration. Thus, ensuring the immutability of blockchain data poses concerns regarding data integrity. These challenges collectively contribute to the need for more market acceptance. Achieving widespread acceptance necessitates collaboration and consensus among various stakeholders, including regulatory bodies, healthcare providers, and patients. The reluctance or skepticism of key players to embrace blockchain technologies introduces a significant barrier. Trust and acceptance, therefore, play pivotal roles in the successful integration of technological innovations within the pharmaceutical sector.

The results of the MICMAC analysis illustrate the drive power dependence matrix, derived from the model, that also provides pharmaceutical professionals and managers with valuable information for a comprehensive understanding of the relative importance, interdependencies, and relationships among these factors. The drive and dependence power diagram provides further insights into their importance and interdependencies. The matrix depicting drive and dependence power reveals three autonomous challenges, namely $Ch_6$, $Ch_{20}$, and $Ch_5$. This indicates that these challenges exert minimal influence on the system, possessing both weak drive and dependent power, indicating less importance within the system. The dependent cluster comprises seven variables: $Ch_{16}$, $Ch_4$, $Ch_{19}$, $Ch_{14}$, $Ch_{17}$, $Ch_{12}$, and $Ch_{18}$. Given their characteristics of weak drive power but strong dependence power, these factors demand a heightened priority in regards to management considerations.

Furthermore, managers should consider the dependence of these factors on elements at other levels within the ISM framework. These findings align with the results of Xu, Chong, and Chi (2023) [146], whose study on modeling blockchain adoption barriers identified, among others, "reluctance to change business processes" and "lack of collaboration and network establishment" as having the lowest driving values, both overall and within the cluster of dependent factors. Ch21, Ch15, Ch10, and Ch13 are positioned within the range of the linkage factors. These factors exhibit both robust driving and substantial dependence power. Any minor intervention undertaken on these challenges is poised to yield a notable impact on other elements, and they also exhibit a feedback effect on themselves. These factors are crucial in fostering a positive environment conducive to blockchain adoption in the pharmaceutical industry. Finally, $Ch_1$, $Ch_9$, $Ch_3$, $Ch_2$, $Ch_8$, $Ch_7$, and $Ch_{11}$ stand as independent factors, boasting formidable drive power but exhibiting

limited dependence power. These are pivotal elements acknowledged as the foundational causes influencing all other factors. This result coincides with that of Yadav et al. (2020) [68], Sahebi et al. (2020) [81], and Sharma et al. (2021) [120], whose conclusion emphasized that the principal barrier to blockchain adoption is the "lack of government regulations and support".

*5.2. Academic Implications*

The SLR highlights numerous studies addressing challenges associated with blockchain adoption, encompassing perspectives tailored to specific industries and those more universal. However, only a limited number of studies examine the interrelationships among these challenges. Prior studies failed assess the challenges in regards to blockchain adoption with the specific goal of improving the sustainability performance of the pharmaceutical industry. This study pioneers a comprehensive examination of challenges of blockchain adoption within the pharmaceutical industry, contributing to a nuanced understanding of associated challenges and their interconnections for further exploration. Acknowledging the impracticality of addressing all challenges simultaneously, the research identified critical factors through an SLR and expert feedback. They subsequently utilized an integrated decision framework, incorporating FISM and MICMAC analyses, to discern the interdependencies, driving, and dependent powers of the challenges associated with adopting blockchain technology in the pharmaceutical industry. Serving as an intriguing first step in research progression, this study encourages scholars to focus on overcoming challenges to blockchain adoption in the pharmaceutical industry in future studies. Notably, this research establishes a foundational framework for subsequent empirical analyses to explore determinants of blockchain technology adoption in the pharmaceutical sector.

*5.3. Managerial Implications*

Vigilant monitoring of crucial variables is essential to foster a conducive environment for effective blockchain adoption in the pharmaceutical industry, enhancing sustainability performance. The model elucidates the intricate interconnections and mutual influences among diverse factors impacting blockchain adoption in the pharmaceutical industry. Evolving challenges in the industry, driven by shifts in market dynamics and patient demands, underscore the growing need for enhanced supply chain visibility and more effective recall processes [147,148]. Considering these challenges, this research scrutinizes pivotal factors influencing blockchain adoption in the pharmaceutical industry, offering a revamped model to guide management in successfully navigating these factors.

Functioning as a disruptive and innovative technology, blockchain has the potential to enhance trust in relationships, ensure secure payments, streamline processes, minimize transaction costs, and improve traceability. Nevertheless, numerous technical, environmental, and organizational challenges persist, as emphasized by the findings of this study. The results underscore that $Ch_1$, $Ch_3$, $Ch_6$, $Ch_{10}$, $Ch_{13}$, and $Ch_{21}$ stand out as the most critical barriers, suggesting that the pharmaceutical industry is currently inadequately equipped to adopt blockchain technology. Top-level management and policymakers in the sector must take decisive and effective actions, demonstrating their commitment to minimizing these barriers. These barriers exhibit significant driving power, acting as fundamental causes for the emergence of other obstacles. Therefore, management and policymakers should act to progressively diminish barriers with substantial driving power to attain the desired objectives in the pharmaceutical sector.

The "high costs of blockchain investments" challenge in the pharmaceutical industry stems from initial implementation expenses, regulatory compliance, and the need for employee training. To address this, collaborative industry efforts, government incentives, and technological solutions, like blockchain-as-a-service, can help offset costs. Furthermore, artificial intelligence (AI) can assist in ensuring that blockchain applications comply with existing regulations by monitoring transactions and flagging those that might violate regulatory standards [149]. A phased implementation approach allows companies to

prioritize key areas and demonstrate blockchain's value gradually, managing expenses more effectively. Overcoming the financial challenge in blockchain adoption requires securing strong leadership support. Total management commitment makes justifying costs easier, hindering resource allocation and integration. To address this, organizations should educate leaders on blockchain benefits, demonstrate clear ROI, initiate small-scale pilots, align initiatives with organizational goals, create phased roadmaps, foster cross-functional collaboration, maintain transparent communication, and develop robust risk management strategies. AI can also support this by assisting in creating and auditing smart contracts by checking for errors and vulnerabilities, reducing the costs and risks associated with manual audits [150]. Building confidence in the technology's potential and showcasing incremental successes are critical to developing the commitment for successful blockchain integration. Ensuring the immutability of blockchain data poses concerns about data integrity. Solutions involve employing advanced consensus mechanisms and periodic audits, as well as adopting hybrid models that balance transparency and flexibility. Collaboration within the industry can establish standards for immutable data while allowing for occasional adjustments. Overcoming skepticism and fostering trust in blockchain technology requires strategic communication, education initiatives, industry partnerships, and advocacy to promote its benefits and showcase successful use cases. Fostering a culture of experimentation and showcasing tangible outcomes contributes to broader acceptance. It requires strategic communication and education initiatives. Industry partnerships and advocacy can promote the benefits of blockchain technology, demonstrating successful use cases. The need for more professionals well-versed in blockchain technology poses a significant challenge. Implementing blockchain technology requires specialized knowledge in cryptography, distributed ledger technology, and smart contracts. Addressing the need for more technology expertise involves investing in training programs, collaborations with educational institutions, and encouraging a cross-disciplinary approach. Companies can also leverage external expertise through partnerships and consultancy to bridge the skills gap during the initial phases of blockchain implementation. Addressing widespread unfamiliarity with blockchain technology and its advantages, as well as its potential, requires a comprehensive approach. Initiatives such as educational programs, collaboration with industry associations, engagement with regulatory bodies, and public awareness campaigns, along with accessible online resources and practical applications, aim to create an informed and supportive environment for blockchain adoption.

## 6. Conclusions and Limitations

### 6.1. Conclusions

The benefits of blockchain technology and its applicability across various industries have prompted companies to integrate blockchain technology into their operations. The existing literature on the challenges associated with blockchain adoption has examined both industry-specific and general perspectives. However, there is a gap in the literature concerning the analysis of challenges specific to blockchain adoption in the pharmaceutical industry. This study conducted a preliminary investigation into the significant challenges hindering the adoption of blockchain technology in the pharmaceutical sector to enhance sustainability performance. by performing an SLR and obtaining feedback from 18 experts, 21 key barriers were identified and analyzed. The FISM analysis was employed to construct a hierarchical model and determine relationships among the factors. A MICMAC analysis grouped factors into four clusters to assess the driving and dependence power of the barriers. The research findings reveal that the 21 factors are divided into five levels, with $Ch_1$, $Ch_3$, $Ch_6$, $Ch_{10}$, $Ch_{13}$, and $Ch_{21}$ identified as the primary challenges to blockchain adoption in the pharmaceutical industry. This research is a valuable preliminary study, establishing the groundwork for future research. The integrated FISM-MICMAC approach offers academicians and industrialists a comprehensive overview of the challenges associated with blockchain adoption in the pharmaceutical industry. The findings provide essential guidance for practitioners in the pharmaceutical industry and government policy-

makers, particularly in selecting potential solutions to address the identified challenges to blockchain adoption, ensuring long-term success and competitiveness in the market. Consequently, this proposed FISM model guides business managers in devising operational strategies to address challenges related to blockchain adoption. Before adopting blockchain in their firms, management must clearly understand the hierarchy of factors. Subsequently, organizations can frame competitive strategies based on the driving and dependence power of various factors.

*6.2. Limitations*

This study has several limitations. One significant constraint is its reliance on subjective judgments from the expert group to establish interdependencies among critical challenges to blockchain adoption, potentially introducing personal bias into the results. Future research should seek to incorporate multiple stakeholder perspectives to assess barriers and compare their similarities and differences, enhancing the objectivity of the decision-making process for policymakers and planners [128,151]. Additionally, online surveys limit access to detailed sample information, raising concerns about the accuracy of the demographic data provided by the participants. Therefore, future studies must employ alternative methodologies and diverse expert panels to ensure robust conclusions.

Additionally, a conceptual model involving different factors influencing blockchain adoption in the pharmaceutical industry has been constructed, incorporating insights from a literature review, inputs gathered through discussions with experts in the relevant field, and findings from the survey. It is important to acknowledge that the model may deviate from real-world scenarios, and the relationships between various factors might differ from those depicted in the derived model. This discrepancy arises because the FISM methodology employed does not quantify the impact of each variable. Moreover, further empirical studies are necessary to gain deeper insights into critical challenges. Structural equation modeling could validate cause–effect relationships, the game theory might be utilized to assign objective weights to criteria for more reasonable results, and adaptive neuro-fuzzy inference systems could be applied to quantitatively prioritize identified challenges. Also, comprehensive and longitudinal studies are warranted to explore whether the prominence of the identified factors and their relationships varies across countries and cultures and to assess how the evolution of these challenges impacts the advancement of new technologies in the pharmaceutical industry.

**Supplementary Materials:** The following supporting information can be downloaded at: https://www.mdpi.com/article/10.3390/su16083102/s1, Table S1: Reviewed literature on challenges of blockchain adoption.

**Funding:** This research received no external funding.

**Institutional Review Board Statement:** Not applicable.

**Informed Consent Statement:** Informed consent was obtained from all subjects involved in the study.

**Data Availability Statement:** Only simulated data is applied, and real-world operational data is unavailable.

**Conflicts of Interest:** The author declares no conflicts of interest.

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
