# Peer review of "Addressing Challenges: Adopting Blockchain Technology in the Pharmaceutical Industry for Enhanced Sustainability"

_sustainability, doi:10.3390/su16083102_

Round 1

Reviewer 1 Report

Comments and Suggestions for Authors

Contributions of the paper :

  • Identifies and understands the challenges hindering the adoption of blockchain in the pharmaceutical sector for improving sustainability performance.
  • Establishes the interdependencies among these challenges using methodologies like Fuzzy Interpretive Structural Modeling (FISM) and Cross-Impact Matrix Multiplication Applied to Classification (MICMAC).
  • Provides a structured framework for improved decision-making and strategic planning in the pharmaceutical industry.

Comments/Suggestions:

  1. Provide a more detailed explanation of the methodologies used, such as Fuzzy Interpretive Structural Modeling (FISM) and Cross-Impact Matrix Multiplication Applied to Classification (MICMAC), to enhance the understanding of the research process.
  2. Include a clear definition of sustainability indicators across environmental, social, and economic pillars in the pharmaceutical industry to establish a common understanding for readers.
  3. Expand on the expert opinions obtained during the refinement of challenges to provide insights into the specific perspectives and experiences of industry professionals.
  4. Include a discussion on the potential benefits and drawbacks of adopting blockchain technology in the pharmaceutical industry for sustainability enhancement to provide a balanced view.
  5. Provide more information on the sample size and characteristics of the experts interviewed to ensure transparency and reliability of the findings.
  6. Include case studies or real-world examples of organizations in the pharmaceutical industry that have successfully adopted blockchain technology for sustainability enhancement to illustrate the practical applications.
  7. Consider including a section on potential solutions or strategies to address the identified challenges, providing practical recommendations for organizations in the pharmaceutical industry.
  8. Expand the discussion on the implications of the research findings, particularly in terms of their impact on brand image, company perception, and consumer value, to provide a deeper understanding of the potential outcomes. 
  9. The authors are invited to include a dedicated paragraph or section that delves into the promising realm of combining blockchain and AI, unraveling the novel possibilities and implications arising from the convergence of these powerful technologies.
  10. For this purpose, the authors may include the following interesting references (and others):
    a. https://www.mdpi.com/2073-431X/12/5/107
    b. https://ieeexplore.ieee.org/abstract/document/8592662
Comments on the Quality of English Language

Can be improved.

Reviewer 2 Report

Comments and Suggestions for Authors

Addressing Challenges: Adopting Blockchain in the Pharmaceutical Industry for Enhanced Sustainability” uses FISM and MICMAC while analysing literature and expert opinions regarding the use of blockchain in the pharmaceutical industry. The article is well written and makes a valuable contribution both through the data and thorough discussions on the implications for academia and industry. The paper provides an excellent and comprehensive literature review and Table 1, which details the identified challenges, will be an important resource for future research.  While the substance of the paper is sound, there are, however, a few minor things to be addressed.

Starting a sentence with “So” is too informal for an academic paper (see ln 59).

In the introduction the paper uses the future tense (“will be identified” etc).  This sounds more like a proposal than the introduction of a finished study.

In Table 2, it is not clear if the experience is total professional experience or experience in the detailed position related to the pharmaceutical industry.

Reviewer 3 Report

Comments and Suggestions for Authors

1. In the introduction, the authors present few examples of the application of blockchain in the pharmaceutical industry. I recommend that they can add some real cases to support the argument.

2. Obviously, three primary streams of research are surveyed in literature review: research on Impact of blockchain and sustainability, research on Sustainability and blockchain in the pharmaceutical industry, and research on employ a Systematic Literature Review (SLR) methodology, the existing body of 175 literature was systematically explored. In this section, only the relevant work of existing literature is described. Please note that literature review are not just a simple pile and list of literature, focusing on the differences between this study and the existing literature to highlight the actual contribution. In addition,  Yu S., Wu C., Xu C. The optimal pricing in blockchain-enabled enterprise operation considering privacy 1.  In addition, lack of recent references, especially in 2023, related to blockchain. For

example, [1] Yu S., Wu C., Xu C. The optimal pricing in blockchain-enabled enterprise operation considering privacy attitude and privacy protection. Asia-Pacific Journal of Operational Research, 2023, 40(4), 2340002,

[2] Iyengar, G., Saleh, F., Sethuraman, J., and Wang, W. Economics of permissioned blockchain adoption. Management Science, 2023, 69(6): 3415-3436.

3. Why use methods Fuzzy Interpretive Structural Modeling and Matrix Multiplication Applied to Classification? What are the advantages of these two methods over other methods for this article?

4. What are the main innovations of the article compared to the content studied in other literature?

5. Detailed information about the sample could not be obtained using online survey methods, and it was not possible to determine whether participants accurately provided demographic or characteristic information. Are the conclusions drawn from the online survey data valid? Can data obtained through other means, and panels experts of varying degrees, reach the same conclusions? I suggested that the author can add this section to make the content of the article more complete.

6. The language of the paper is not on publication level and needs thorough revisions. If the authors are not able to bring the paper's language up to publication level by themselves. I recommend that they involve a native speaker or a professional language polishing agency.

Comments on the Quality of English Language

Minor editing of English language required

Round 2

Reviewer 1 Report

Comments and Suggestions for Authors

The authors considered my comments and suggestions. Good luck.

Comments on the Quality of English Language

A final proofread would be useful.

Reviewer 3 Report

Comments and Suggestions for Authors

Minor editing of English language required.

Comments on the Quality of English Language

Minor editing of English language required.